# A Holistic Approach to Identify and Classify COVID-19 from Chest Radiographs, ECG, and CT-Scan Images Using ShuffleNet Convolutional Neural Network

**DOI:** 10.3390/diagnostics13010162

**Published:** 2023-01-03

**Authors:** Naeem Ullah, Javed Ali Khan, Shaker El-Sappagh, Nora El-Rashidy, Mohammad Sohail Khan

**Affiliations:** 1Department of Software Engineering, University of Engineering and Technology Taxila, Taxila 47050, Pakistan; 2Department of Software Engineering, University of Science and Technology Bannu, Bannu 28100, Pakistan; 3Faculty of Computer Science and Engineering, Galala University, Suez 435611, Egypt; 4Information Systems Department, Faculty of Computers and Artificial Intelligence, Benha University, Banha 13518, Egypt; 5Department of Machine Learning and Information Retrieval, Faculty of Artificial Intelligence, Kafrelsheiksh University, Kafr Elsheikh 33516, Egypt; 6Department of Computer Software Engineering, University of Engineering and Technology Mardan, Mardan 23200, Pakistan

**Keywords:** chest radiographs, convolutional neural networks, COVID-19, classification, CT scans, detection, ECG Trace Images, medical imaging, ShuffleNet

## Abstract

Early and precise COVID-19 identification and analysis are pivotal in reducing the spread of COVID-19. Medical imaging techniques, such as chest X-ray or chest radiographs, computed tomography (CT) scan, and electrocardiogram (ECG) trace images are the most widely known for early discovery and analysis of the coronavirus disease (COVID-19). Deep learning (DL) frameworks for identifying COVID-19 positive patients in the literature are limited to one data format, either ECG or chest radiograph images. Moreover, using several data types to recover abnormal patterns caused by COVID-19 could potentially provide more information and restrict the spread of the virus. This study presents an effective COVID-19 detection and classification approach using the Shufflenet CNN by employing three types of images, i.e., chest radiograph, CT-scan, and ECG-trace images. For this purpose, we performed extensive classification experiments with the proposed approach using each type of image. With the chest radiograph dataset, we performed three classification experiments at different levels of granularity, i.e., binary, three-class, and four-class classifications. In addition, we performed a binary classification experiment with the proposed approach by classifying CT-scan images into COVID-positive and normal. Finally, utilizing the ECG-trace images, we conducted three experiments at different levels of granularity, i.e., binary, three-class, and five-class classifications. We evaluated the proposed approach with the baseline COVID-19 Radiography Database, SARS-CoV-2 CT-scan, and ECG images dataset of cardiac and COVID-19 patients. The average accuracy of 99.98% for COVID-19 detection in the three-class classification scheme using chest radiographs, optimal accuracy of 100% for COVID-19 detection using CT scans, and average accuracy of 99.37% for five-class classification scheme using ECG trace images have proved the efficacy of our proposed method over the contemporary methods. The optimal accuracy of 100% for COVID-19 detection using CT scans and the accuracy gain of 1.54% (in the case of five-class classification using ECG trace images) from the previous approach, which utilized ECG images for the first time, has a major contribution to improving the COVID-19 prediction rate in early stages. Experimental findings demonstrate that the proposed framework outperforms contemporary models. For example, the proposed approach outperforms state-of-the-art DL approaches, such as Squeezenet, Alexnet, and Darknet19, by achieving the accuracy of 99.98 (proposed method), 98.29, 98.50, and 99.67, respectively.

## 1. Introduction

Last year, a global epidemic was triggered by the most recent coronavirus named COVID-19. The first outbreak of COVID-19, which was transmitted to humans by bats, was observed in Wuhan, Hubei Province, China, in December 2019 [1]. This condition, which can be fatal, is caused by the severe acute respiratory syndrome coronavirus 2 (SARS-CoV-2). The COVID-19 virus is fatal because of its fast transmission. This virus can spread through the air and physical contact, such as a handshake with a COVID-19 patient [2]. This virus transfers from one animal to another and from animals to humans. Coughing, fever, and shortness of breath are all frequent symptoms of COVID-19 [3]. In most cases, the virus damages the human lungs, causing pneumonia in severe cases. The World Health Organization (WHO) claims that 649,038,437 verified cases of COVID-19 have been noted worldwide to date, with 6,645,812 deaths [4]. According to WHO 268,252,496; 184,161,028; 60,719,433; and 9,431,508 cases of COVID-19 have been reported in Europe, America, Southeast Asia, and Africa, respectively. 

Existing COVID-19 detection tests are slow and usually take a few hours to obtain the required results. In most cases, in medical research, the polymerase chain reaction (PCR) test is used [5]. Unfortunately, because the number of cases is continually growing, doing enough PCR testing has become almost impossible due to the time, shortage of medical resources, and cost involved [6]. As the need for COVID-19 testing has expanded, laboratory professionals have run across a growing number of obstacles, doubts, and, in some cases, disputes. As a result, there is a compelling need to create alternative testing (computerized COVID-19 detection) technologies that can consistently detect the virus in a short period of time to recognize infected people and quarantine or isolate them promptly.

To reliably and automatically detect (identify or predict) COVID-19 in its early phases, various medical imaging techniques, such as chest radiographs, ECG trace images, and CT-scans have been used [7]. Chest radiographs, often known as X-rays, are images of the inside of the chest that are utilized to examine chest problems [1]. ECG trace images are line graphs that depict variations in the heart’s electrical behavior over time [7]. On the other hand, a chest CT scan employs an X-ray scanner to produce a sequence of high-resolution images of locations inside the chest [8]. Medical professionals value and prefer chest radiograph images more because they can be easily accessed from radiology departments. Chest radiograph images, according to radiologists, aid in the clear understanding of chest pathology [2]. Additionally, the ECG trace images can easily be taken and gathered by mobile phones and are quickly accessible technologies in nations with limited resources and budgets. Therefore, the X-ray modality [9] and ECG images were the first low-cost and low-risk tools for analyzing COVID-19. The X-ray technique is widely utilized for pneumonia diagnosis [10]. Chest radiography exposes people to less radiation than magnetic resonance imaging (MRI) and computed tomography (CT) [11].

Several research studies have shown that COVID-19 may be detected on chest radiographs with radiologist-level accuracy using traditional machine learning (ML) or DL-based CAD systems, which can be employed in medical practice [12,13,14]. Compared to traditional ML techniques, DL techniques make use of unstructured data, automatically extract more robust features, and produce more accurate results. At the beginning of 2022, various studies have been undertaken to construct automated DL approaches for reliable recovery of COVID-19 [15,16]. Most of these studies used convolutional neural networks (CNNs) to classify COVID-19-infected or normal chest radiograph images.

Existing works on COVID-19 detection have some limitations: The majority of studies relied on datasets with fewer images (training data); more specifically, limited ECG trace images data is available for COVID-19 detection and as a result the model is not generalized, and the model may have overfitted the training samples. Furthermore, the backpropagation algorithm employed in CNN training is often very slow and requires the tuning of different hyperparameters. In most studies, novel CNN algorithms are utilized as potential classifiers, but the CNN algorithm has some pre-existing limitations. For example, when there are any imbalanced classes in the dataset, it can be overfitted. The majority of previous algorithms were designed to train on either chest radiographs, CT scans, or ECG trace images, i.e., only one or two types of data. However, the DL algorithm trained on different data types (images) can extract more detailed information (most reliable deep feature) for classification. Most of the studies perform either binary or multiclass classification of images. Additionally, other state-of-the-art CNN frameworks have not been compared separately in many studies.

To address the above-mentioned limitations of existing approaches in this area, we propose an effective single DL-based framework for chest radiograph, CT scan, and ECG trace images to identify COVID-19 positive cases. The proposed framework utilizes filter-based feature extraction, which can be useful in attaining the greatest classification performance. The main contributions and novelty of this work are as follows: (1) We used ShuffleNet, a transfer learning (TL) based framework for chest radiographs, CT scans, and ECG trace images to detect and identify COVID-19 positive cases; (2) We examined and evaluated the classification performance of the ShuffleNet pre-trained DL framework in terms of their ability to identify COVID-19 using three different types of images data; (3) Using multiple data sources, we utilized the same framework for binary and multiclass classification to detect COVID-19 positive instances (COVID-19 infected individuals); (4) To assess the suggested model’s efficacy, we compared its performance to that of existing state-of-the-art DL frameworks on the same dataset and with the same experimental setup; and (5) We assessed the classification performance of the suggested approach on widely accessible standard datasets, such as an ECG image dataset of cardiac and COVID-19 patients, a SARS-CoV-2 CT scan dataset, and a standard COVID-19 Radiography Database.

The following is how the rest of the paper is organized: Section 2 describes the related work. In Section 3, we offered the motivation for the proposed work as well as an explanation of it. Section 4 detailed the datasets, evaluation measures, and experimental outcomes in depth. Section 5 completes the paper.

## 2. Related Work

Although COVID-19 has only recently begun to spread, researchers have completed many research projects in a short period. Various ML, hybrid, and DL algorithms have been proposed to categorize COVID-19 images, which is the present problem. Several works have been briefly discussed and summarized.

In [17], for COVID-19 identification and detection, the authors used multi-level thresholding and SVM. They analyzed the patient’s lung radiograph and utilized a median filter to improve the contrast of the input radiographs. The support vector machine (SVM) was then utilized to differentiate between diseased and normal lungs. In [8], COVID-19 was automatically identified from CT scans utilizing a variety of ML methods. Several feature extraction algorithms, including the grey-level size zone matrix (GLSZM), grey-level co-occurrence matrix (GLCM), and grey local directional pattern (GLDP), and the discrete wavelet transform (DWT) technique were used to improve the performance. The classification of extracted features was done using SVM with two-fold, five-fold, and ten-fold cross-validations. The best accuracy was attained using GLSZM and ten-fold cross-validation.

In [18], the authors utilized chest radiograph images to identify coronavirus infected patients by utilizing a deep feature and SVM-based approach. Deep features from the Convolutional neural networks (CNN) fully connected layers are retrieved and supplied to SVM for image classification. For COVID-19 identification, ResNet50 plus SVM attained the best accuracy, sensitivity, FPR, and F1 scores. In [19], the authors used the CNN models to extract features and the SVM as a classification method to categorize radiograph images into normal, COVID-19 positive, and pneumonia. They attained the highest average accuracy through ResNet50, ResNet18, ResNet101, and GoogleNet. Furthermore, different optimization algorithms were produced to improve the performance of machine learning algorithms [20,21,22,23,24].

Traditional ML techniques perform poorly compared to DL techniques since they rely largely on manual feature extraction, whereas, DL approaches automatically extract more robust deep features, and deliver more accurate results than typical ML techniques. In [25], the authors used a truncated Inception network to differentiate COVID-19 and normal images. In [26], the authors used the DarkNet architecture to construct a binary classifier that detects COVID-19 and normal chest radiograph images and a multi-class classifier that detects COVID-19 positive, pneumonia, and normal. On the ImageNet dataset, the authors of [27] employed the pre-trained Xception framework. The authors of [28] utilized MobileNet to train a framework from scrape and extract features for the classification challenge. The authors [29] employed the Bayesian optimization method to tune the SqueezeNet network on the COVID-19 diagnostic. Furthermore, different optimization algorithms were produced to improve the performance of machine learning algorithms [30,31,32,33].

Recently, when CT scans and clinical observation histories are found, the artificial neural network effectively diagnoses a coronavirus patient [34]. A previous study used CNN to diagnose COVID-19 illness from a chest radiograph using the inception network [35]. In [36], the author proposed a new ensemble-based technique to efficiently and accurately identify COVID-19 by utilizing CT scans. They employed a TL to classify clustered images of lung lobes using pre-trained DL frameworks, such as Xception, ResNet, VGG, and integrated them into an ensemble framework. In [37], the authors used 11 DL frameworks to categorize chest radiographs as normal, COVID-19, or pneumonia. They evaluated three distinct modifications to change the frameworks for the classification task by adding some additional layers. The EfficientNetB4 and Xception-based frameworks attained the best classification performance. In [38], the authors purposed a CNN to identify COVID-19 using chest radiographs. Deep feature extraction has been done using pre-trained CNN frameworks, including VGG16, InceptionV3, MobileNetV2, and ResNet50. InceptionV3 obtained the highest accuracy to detect SARS-CoV-2 from chest radiograph images. More recently, DL approaches on ECG trace images have been explored with promising results. The authors [39] used DL approaches to identify COVID-19 and other cardiovascular illnesses (CVDs) from ECG trace images. Six deep CNN classifiers were utilized to perform a series of classification experiments. The Densenet201 algorithm outperforms other algorithms in binary and three-class fine-grained classification, whereas InceptionV3 surpasses others in five-class classification.

There are various limitations of the approaches discussed above in the related work. According to our knowledge, the existing methods on COVID-19 detection are unable to achieve improved classification performance. One possible reason might be that research studies have utilized databases with fewer images, and there is a potential that their proposed framework has overfitted the training images. Furthermore, their comparisons are made on one (single) dataset, which might hinder the model’s effectiveness. For example, when only one type of image data is utilized to train and validate the framework, it is not considered a generalized model. As a result, an effective and generalized classification system and a unified model (a model that can detect COVID-19 utilizing several types of image data) are required to solve the drawbacks of current COVID-19 detection methodologies.

## 3. Methodology

The proposed COVID-19 detection and classification mechanism is discussed in depth in this section. The proposed DL system applies the ShuffleNet DL framework to detect COVID-19 and classify the images of three datasets, as shown in Figure 1. The proposed work comprises three main stages: In the first stage, COVID-19 is detected using chest radiograph images, in the second stage, COVID-19 is detected using CT scan images, whereas in the third stage COVID-19 is detected using ECG trace images represented with revise stage in Figure 1. To the input data, we used a built-in MATLAB DL toolbox to train the proposed model. The images in the datasets are different in size. According to the input image requirements of the proposed model, we downsized all of the images to 224 × 224 pixels. For all experiments, the datasets were separated into training and testing datasets. To be more explicit, we utilized 80% of the images for training and 20% of the images for testing. The weights of the pre-trained ShuffleNet network are fine-tuned by freezing the weights of the first layers—that is, the weights of the frozen layers are not adjusted during training—while the fully connected (FC) layers, which map the feature representations extracted by the first layers into the class label information, are fine-tuned. The model’s classification layer and final FC layer are replaced, which was initially planned to generate 1000 separate categories. The proposed framework is used to accomplish both binary and multiclass classification to detect COVID-19-positive instances (COVID-19-infected individuals) using chest radiographs and ECG trace images.

### 3.1. Motivations

COVID-19 has symptoms similar to severe pneumonia, so inspired by the success of DL-based architecture in pneumonia detection [40], we proposed a COVID-19 identification and classification approach based on the ShuffleNet DL model in this work. In addition, inspired by pattern recognition and computer vision techniques, where the same framework is trained for several different classes/objects. We proposed an effective single DL-based framework for chest radiographs, CT scans, and ECG trace images to identify COVID-19 patients. This study aims to develop a unified framework that could be utilized for binary classification (e.g., COVID-19 identification) and multiclass classification (e.g., COVID-19, normal, and other CVDs classification) with greater accuracy using chest radiographs, CT scans, and ECG trace images.

The importance and properties of the proposed method (ShuffleNet) include (1) it requires minimal dataset pre-processing; (2) it saves time (TL-based approach) by eliminating the need to train and validate the model weights from scratch; (3) it reduces the computation cost because the CNNs are often made up of repeating building blocks with the same structure and do not completely account for 1 × 1 convolutions (pointwise convolutions), which need a great deal of complexity. ShuffleNet, on the other hand, reduces the computation cost (to speed up training) by using group convolution on 1 × 1 layers. Furthermore, the model is based on pointwise group convolution and channel shuffle, which decrease computational costs while retaining classification accuracy and are optimized for devices with low computational capacity. Batch normalization (BN) is used after each convolutional layer to standardize the inputs, provide regularization, and reduce generalization error (to improve generalization ability).

### 3.2. ShuffleNet Architecture Details

In the proposed work, we employed the ShuffleNet [41], a highly efficient DL architecture created with mobile devices. Based on our computational resources (hardware), we used shufflenetv1 version of the pre-trained ShuffleNet model to achieve better accuracy at low computational costs. The architecture of the proposed framework is revealed in Figure 2. The proposed model is deeper than standard CNN with 50 learnable layers, i.e., 1 convolution layer and 48 group convolution layers followed by an FC layer. The architecture has a total of 172 layers, including 1 maximum pooling layer, 49 BN layers, 33 relu layers, 4 average pooling layers, a softmax layer, and a classification layer. The framework employs four pooling layers to decrease the overall computational complexity.

The input layer of our model is the initial layer, and it accepts 224 × 224 input images (chest radiograph, CT scan, or ECG trace image) for processing. To generate the feature map, the first convolution layer extracts the feature from the 224 × 224 input image by applying at a time 24 kernels (filters) of size 3 × 3 with a stride of 2 × 2. The output of convolutional layers (feature map) is calculated as:(1)$$s\left(i,j\right)=\left(I\times K\right)\left(i,j\right)=\underset{n}{\sum }\underset{m}{\sum }I\left(m,n\right)K\left(i-m,j-n\right)$$
*s* represents the output feature map, *i* represent the input image, whereas *K* represents the kernel of the current convolutional layer. After applying convolution operations on the input image, the output of size $o=\left(\left(i-k\right)+2p\right)/\left(s+1\right)$ is produced, where *i* represents input, *p* means padding, *k* represents kernel size, and s represent steps.

The ShuffleNet unit with a shift (stride) of 2 × 2 receives the output feature map of the first convolutional layer. The ShuffleNet unit comprises three convolutional operations, i.e., two 1 × 1 pointwise group convolution and 3 × 3 depthwise convolutions. The first pointwise group convolution is followed by BN, relu activation function, channel shuffle operation. Relu activation is used because it is efficient and straightforward. Relu works as follow
(2)$$f\left(x\right)=\left\{\begin{array}{c} 0,\;x<0\\ x,\;x\;\leq 0 \end{array}\right.$$

Relu activates neurons with positive values and deactivates neurons (set neurons to 0) with negative values. The second and third convolution operations, i.e., 3 × 3 depthwise convolutions and 1 × 1 pointwise group convolution, are followed by BN. The model contains a 3-by-3 average pooling on the shortcut paths. The model consists of 16 consecutive ShuffleNet units. The model is made up of 50 layers, each of which provides trainable feature maps. These layers also do feature extraction. These feature maps are submitted to FC, and Soft-max activation is utilized to determine the classification probabilities used by the final classification layer. Equation (3) represents the working of the FC layer.
(3)$$a_{i}=\overset{m\times n-1}{\underset{j=0}{\sum }}w_{ij}\times x_{i}+b_{i}$$
where *i* denotes the index of the FC layer’s output; *n*, *m*, *d*, and *i* denote the height, width, depth, and index of FC layers output. Furthermore, *b* and *w* represent the bias and weights, respectively. The Soft-max layer’s classification probabilities can create up to 1000 separate classes, but we have two, three, four, and five classes in our experiments.

### 3.3. Hyperparameters Settings

We used a trial-and-error approach in which we ran trials with various parameter values to determine the best value for each one. The proposed techniques’ hyperparameters are chosen after some preliminary trials on a smaller dataset. Table 1 illustrates the details of the parameters that are chosen. Our model was trained using stochastic gradient descent (SGD) with a learning rate of 0.001. The proposed framework is trained over 22 epochs to classify radiograph, CT scans, and ECG trace images into binary and multiclass categories.

## 4. Results

In this section, we provide an in-depth discussion of the findings of numerous experimentations designed to evaluate the performance of the proposed approach. This section comprehends further information about the datasets used to assess the classification performance of our approach, specifically the COVID-19 Radiography, SARS-CoV-2 CT, and the ECG images dataset of cardiac and COVID-19 patients.

### 4.1. Research Datasets

For COVID-19 finding through chest radiographs, we utilized the COVID-19 Radiography Database [42,43], which was created by scientists from Bangladesh’s University of Dhaka and Qatar University. This COVID-19 chest radiograph images database was created in collaboration with medical doctors for COVID-19, pneumonia, and normal radiographs. The COVID-19 chest radiograph images database was just released; it includes 3616 chest radiographs of COVID-19 infected persons, 10,192 chest radiographs of healthy people, 6012 Lung Obesity, and 1345 Pneumonia radiographs. The radiographs in this dataset have a resolution of 1024 × 1024 pixels, as shown in Table 2. The images were resized to fit the needs of each model. It is a standard Kaggle dataset that is open to the public. This dataset’s radiograph images are grayscale and have the same dimensions. A few examples from the SIRM dataset are displayed in Figure 3.

We detected COVID-19 by utilizing CT scans from a freely accessible SARS-CoV-2 CT scan dataset [44] comprising 1252 CT scans of COVID-19 patients, with 1230 CT scans of normal people (total of 2482 CT scans). These statistics were gathered from real patients in Brazilian hospitals. Figure 4 displays a few examples from the dataset.

We employed an ECG images dataset of cardiac and COVID-19 patients [45] in this work for COVID-19 identification using ECG, which comprised 1937 unique patient records separated into 5 groups (Normal, COVID-19, RMI, AHB, and MI). All of the images were together utilizing the ECG device ‘EDAN SERIES-3,’ which was deployed in Cardiac Care Units of various health care organizations in Pakistan. Under the observation of qualified medical practitioners with expertise in ECG analysis, 12 lead ECG trace images were recorded and physically analyzed by professors utilizing a telemedicine ECG diagnostic system. The dataset included 859 Normal, 250 COVID-19, 203 RMI, 548 AHB, and 77MI ECG trace images. Figure 5 shows a COVID-19 sample ECG image of the dataset.

### 4.2. Evaluation Metrics

The performance of the proposed framework is validated by computing the following evaluation metrics: Accuracy, Precision, Sensitivity (Recall), specificity, and F1_score. The accuracy of the proposed approach is given by Equation (4), defined as “the number of correctly detected or classified images (COVID-19 or normal) to the total number of sample images”. The precision of the proposed approach is identified as “the number of correctly detected or classified images (COVID-19) to the total number of (COVID-19) positive images detected (correctly or erroneously) by the model”. The recall is calculated as “the number of correctly classified images (COVID-19) to the total number of COVID-19 images in the dataset”. Similarly, specificity is calculated as “the number of correctly detected negative images to the total number of negative (normal) images in dataset”, whereas F1_score combines precisions and recall and calculates the weighted average of both. The equations to estimate these metrics are:(4)$$Accuracy=\left(TN+TP\right)/TS$$
(5)$$Precision=\frac{TP}{TP+FP}$$
(6)$$Sensitivity\;\left(recall\right)=\frac{TP}{TP+FN}$$
(7)$$Specificity=\frac{TN}{TN+FP}$$
(8)$$F1\_score=2\cdot \frac{Precision\times Recall}{Precision+Recall}$$

*TP*, *TS*, *FP*, *TN*, and *FN* stand for true positive, total samples, false positive, true negative, and false negative, respectively.

### 4.3. Experimental Setup

The proposed approach testing and validation are conducted on a machine equipped with an Intel (R) Core (TM) i5-5200U CPU and 8 GB of RAM. To complete the research, we used the R2020a version of MATLAB. Input images were resized according to the input image requirements of models. We utilized 80% of images for training and 20% for validation. The training and testing sets were used in all experiments to train and validate our proposed model and other contemporary models using the identical experimental parameters for COVID-19 detection as those listed in Table 1. A series of experiments were performed to evaluate the classification performance of the proposed framework for binary and multiclass classification utilizing chest radiographs, CT scans, and ECG images.

### 4.4. Performance Evaluation on COVID-19 Detection

In this section, we assessed the performance of the proposed approach on COVID-19 detection. We performed classification experiments using three types of images. With the chest radiographs, we performed two classifications (binary and multiclass-class classification) experiments on the COVID-19 Radiography Database. Also, with CT scans, we performed a binary classification experiment on SARS-COV-2 CT scan dataset. Finally, with ECG images, we conducted two classifications (binary and multiclass-class classification) experiments on ECG Images of Cardiac and COVID-19 Patients.

#### 4.4.1. Performance Evaluation on COVID-19 Detection Using Chest Radiograph Images (Binary Classification)

This experiment aims to validate the performance of the proposed approach for COVID-19 detection using chest radiograph images. For this experiment, we used all the 13,808 chest radiograph images (3616 COVID-19 radiographs and 10,192 normal radiographs) of the dataset. We used 11,047 images (2893 COVID-19 radiographs and 8154 normal radiographs) for training. The remaining 2761 images (723 COVID-19 radiographs and 2038 normal radiographs) were used for testing. The training of the proposed framework took 1262 min and 58 s for COVID-19 detection using chest radiographs. The proposed framework achieved higher TN and TP values, as well as lower FN and FP values at 22 epochs by misclassifying 5 radiographs out of 2761 in the testing phase, as shown in the Table 3. The proposed approach achieved the best accuracy, precision, recall, specificity, and F1-score of 99.82%, 99.72%, 99.59%, 99.90%, and 99.65%. Demonstrating its reliability in COVID-19 detection. The proposed method significantly extracts the more robust features to describe the chest radiograph image for accurate and reliable classification, resulting in these results.

#### 4.4.2. Performance Evaluation on COVID-19 Detection (Multiclass Classification) Using Chest Radiographs

This experiment aims to assess the classification performance of the suggested framework for COVID-19 identification in the case of multiclass classification using chest radiographs. We considered two multiclass classification schemes for this experiment, i.e., three-class classification (COVID-19, normal, and pneumonia) and four-class classification (COVID-19 and Normal, pneumonia, and lung obesity). 

In the first scheme, we used all the 15,153 radiograph images (3616 radiographs of COVID-19, 1345 Viral Pneumonia radiographs, and 10,192 radiographs of healthy individuals) of the database. The 12,123 radiographs (2893 radiographs of COVID-19, 1076 Viral Pneumonia radiographs, and 8154 radiographs of healthy individuals) were used for training, whereas the remaining 3030 radiographs (723 radiographs of COVID-19 patients, 269 Viral Pneumonia radiographs, and 2038 radiographs of healthy individuals) were used for testing. The training of the proposed model in the first scheme took 1460 min and 24 s for COVID-19 detection. The proposed framework achieved higher TN and TP values and lower FN and FP values at 22 epochs by misclassifying only one COVID-19 radiograph image out of 3030 in the testing phase, as shown in Table 4. The proposed technique attained the best accuracy, precision, recall, specificity, and F1-score of 99.98%, 100%, 100%, 99.97%, and 100%, respectively, demonstrating its effectiveness in COVID-19 three-class classification.

In the second scheme, we used all the 21,165 radiograph images (3616 images of COVID-19 patients, 1345 Viral Pneumonia radiographs, 6012 Lung obesity, and 10,192 radiographs of normal individuals) of the dataset. We used 16,933 radiographs (2893 images of COVID-19 patients, 1076 Viral Pneumonia radiographs, 8154 radiographs of normal individuals, and 4810 Lung obesity) for training. The remaining 4232 radiographs (723 images of COVID-19 patients, 269 Viral Pneumonia radiographs, 2038 radiographs of normal individuals, and 1202 Lung obesity) were used for testing. The training of the proposed framework in the second scheme took 2108 min and 24 s. The proposed framework achieved the best results at 22 epochs by misclassifying “6” radiographs out of 4232 in the testing phase, as shown in Table 5. The proposed method attained the highest accuracy, precision, recall, specificity, and F1-score of 99.78%, 99.75%, 99.75%, 99.93%, 99.75%, respectively, for demonstrating the effectiveness of the proposed framework in COVID-19 four-class classification, as elaborated in Table 6.

With an average accuracy, precision, recall, specificity, and F1-score of 99.98%, 100%, 100%, 99.97%, and 100%, respectively, for three-class classification utilizing chest radiograph images, it is verified that our model performed best in the first scheme in terms of accuracy and specificity when compared to the second classification schemes. To assess the proposed approach training performance, we have shown accuracy and loss in Figure 6, elaborating that accuracy and loss after epoch “6” almost remain the same, which means we can obtain satisfactory results even at lower classification epochs. It is to be noted that the model attained high precision and recall and an F1-score of 100% in the first classification scheme, i.e., three class classifications. The proposed model earned the same precision, recall, and F1-score of 100% and 99.75% in the first and second schemes. It is clear from the results of both schemes that the proposed model achieved satisfactory results greater than 99% in terms of all performance metrics. The proposed model adequately extracts the most important and reliable features to represent the radiograph image for accurate and reliable classification, resulting in these findings. The proposed method effectively extracts the most discriminating features to characterize the radiograph image for accurate and reliable classification, resulting in these findings.

#### 4.4.3. Performance Evaluation on COVID-19 Detection Using CT Scans (Binary Classification)

This experiment aims to use COVID-19 CT scans to assess the performance of the suggested approach for COVID-19 detection. We used all the 2482 COVID-19 CT scans (1252 COVID-19 CT scans and 1230 healthy CT scans) of the dataset for this experiment. We used 1986 CT scans (1002 COVID-19 and 984 healthy images) to train the model. Whereas the rest of the 496 images (250 COVID-19 images and 246 normal images) were used for testing. For COVID-19 detection utilizing CT-scans, the proposed framework required 329 min of training. The proposed framework achieved higher optimal TN and TP values of 246 and 250 at 22 epochs by misclassifying 0 CT scans out of 496 in the testing phase, as shown in Table 7.

To assess the training performance of our approach, we have shown the model accuracy and loss in Figure 7, which shows that accuracy and loss after epoch 14 almost remain the same that means we obtain satisfactory results even at lower epochs (15 epochs). The proposed method yielded the best accuracy, precision, recall, specificity, and F1-score of 100%. Although the CT scan images contain noise (such as darkness and low contrast), the proposed method can still extract the distinctive features from the CT scans and detect COVID-19 with optimal accuracy. The proposed framework effectively extracts the more robust features to describe the CT scan image for accurate and reliable classification.

#### 4.4.4. Performance Evaluation on COVID-19 Detection Using ECG Images (Binary Classification)

This experiment aims to evaluate the performance of the proposed approach for COVID-19 identification using ECG trace images. For this experiment, we considered four binary classification schemes, i.e., we performed binary classification of COVID-19 and Normal, COVID-19 and MI, COVID-19 and AHB, and finally COVID-19 and RMI images of the dataset named ECG Images dataset. In the first case, we used all the 1009 ECG trace images (250 COVID-19 images and 859 healthy images) of the dataset. In contrast, 807 images (200 COVID-19 ECG images and 687 normal images) are utilized for training and the rest of the 202 images (50 COVID-19 ECG images and 172 healthy images) for testing. In the second case, we used all the dataset’s 327 ECG images (250 COVID-19 ECG images and 77 ECG images of MI patients). Similarly, 261 images (200 COVID-19 ECG images and 61 MI images) were utilized for training and the rest of the 66 images (50 COVID-19 ECG images and 16 images of MI) for testing. In the third case, we used all the dataset’s 798 ECG images (250 ECG images of COVID-19 and 548 ECG images of AHB individuals). At the same time, 638 images (200 COVID-19 ECG images and 438 AHB images) were utilized for training and the remaining 160 images (50 COVID-19 ECG images and 110 AHB images) for testing. In the fourth case, we utilized all of the dataset’s 453 ECG images (250 ECG images of COVID-19 patients and 203 ECG images of RMI), whereas 362 images (200 COVID-19 ECG images and 162 RMI images) were utilized for training and the remaining 91 images (50 COVID-19 ECG images and 41 RMI images) for validation.

Table 8 summarizes the experimental findings from the four DL experiments. Compared to all three remaining classification schemes, the proposed model attained the highest accuracy and specificity results in the first scheme, with an accuracy of 99.10% and a specificity of 98.85% for COVID-19 detection using ECG trace images. In the third classification scheme, COVID-19 and ABH classification, the proposed model has a second-best accuracy of 98.74%, but the model has the lowest accuracy of 96.92% in the second classification scheme. It is to be highlighted that all comparable models had an accuracy of greater than 95%. In all binary classification schemes, the proposed method achieved the same precision, recall, and F1-score of 96.00%, 1.00%, and 97.96%, respectively. The proposed framework effectively extracts the most discriminating features to represent the ECG trace image for accurate and reliable classification, resulting in these findings.

#### 4.4.5. Performance Evaluation on COVID-19 Detection Using ECG Images (Multiclass Classification)

This experiment aims to measure the performance of the proposed framework for COVID-19 identification using ECG trace images. We considered two multiclass classification schemes for this experiment, i.e., three-class classification (COVID-19, normal, and other CVDs) and five-class classification (COVID-19 and Normal, MI, ABH, RMI). In the first scheme, we used all the 1937 ECG images (250 ECG images of COVID-19 patients, 859 ECG images of healthy individuals, and 829 ECG images of other CVDs) of the dataset, whereas, 1550 images (200 COVID-19 ECG images and 687 ECG images of healthy individuals, and 663 ECG images of other CVDs) were used for training, and the remaining 387 images (50 COVID-19 ECG images and 172 normal images, and 165 ECG images of other CVDs) for testing. The second scheme used all the 1937 ECG images (250 COVID-19, 859 normal, 77 MI, 548 AHB, and 203 RMI) of the dataset. In contrast, 1550 images (200 COVID-19, 687 normal, 62 MI, 439 AHB, and 162 RMI) were used for training, the rest of the 387 images (50 COVID-19, 172 normal, 15 MI, 109 AHB, and 41 RMI) for validation. The proposed framework achieved higher results (in the first classification scheme) at 22 epochs by misclassifying only 6 ECG images out of 387 in the testing phase, as shown in Table 9. The suggested framework achieved higher results (in the second classification scheme) at 22 epochs by misclassifying only 6 ECG images out of 387 in the testing phase, as shown in Table 10. The accuracy and loss performance of the proposed approach training is shown in Figure 8. We obtained the maximum accuracy and minimum loss at epochs 22. The results obtained from the classification experiments are depicted in Table 11. The experimental findings show that the proposed model attained the best results in the second scheme in terms of average accuracy and specificity compared to the first classification schemes by achieving an accuracy of 99.37% and a specificity of 99.12% for multiclass classification using ECG trace images. It is important to consider that the model achieved high precision and recall of 98.00% and 99.0% in the first classification scheme, i.e., three-class classification. These results are because the proposed model effectively extracts the most important and reliable features to represent the ECG trace image for accurate and reliable classification.

### 4.5. Comparison with Hybrid Approaches (Sufflenet + SVM) 

In this section, we conducted an experiment using classical decision-making methods (SVM) instead of SoftMax structures in ShuffleNet. This experiment aimed to validate that using SVM instead of SoftMax could reduce the computational complexity and compare the performance of the proposed method with the hybrid method (Shufflenet and SVM). Hence, we designed a hybrid approach in which we used the ShuffleNet for in-depth features extraction and used these features as inputs to train SVM with linear kernel. Dataset images are resized according to image input requirements of ShuffleNet by using augmented image data stores before inserting them into the network for feature extraction. We applied activations on the last global average pooling layer (a deeper layer) to extract high-level features. According to the results, we achieved maximum results in three classification schemes, i.e., a three-class classification scheme using chest radiographs, binary classification using CT scans, and a five-class classification scheme using ECG images. Therefore, we used these three classification schemes in this experiment. The classification results of deep features and the SVM approach are presented in Table 12. This experiment showed that deep features of ShuffleNet and the SVM approach achieved lower accuracy results than the proposed approach. However, this approach is more efficient than the proposed approach in which we used SoftMax. Based on this experiment, we can conclude that using SVM as a classifier instead of a SoftMax classifier can reduce the computational complexity to a greater extent but affect the performance of the model.

### 4.6. COVID-19 Detection Comparison with State-of-the-Art DL Models

In this section, we performed experiments to confirm the effectiveness of the proposed method for COVID-19 detection over the contemporary DL models. According to our results, we achieved maximum results in three classification schemes, i.e., a three-class classification scheme using chest radiographs, binary classification using CT scans, and a five-class classification scheme using ECG images. So, we compared the performance of our method (in above classification schemes) with different contemporary models.

#### 4.6.1. COVID-19 Three-Class Classification Using Chest Radiographs Comparison with State-of-the-Art Deep Learning Models

This experiment evaluates the usefulness of the proposed framework for COVID-19 three-class classification over the contemporary DL frameworks using chest radiograph images. The proposed method attained the highest accuracy in the three-class classification scheme using chest radiograph images. We compared the proposed method with contemporary models just for COVID-19 three-class classification rather than binary and four-class classification. For this purpose, we compared the classification performance of our framework with three pre-trained DL models, i.e., Squeezenet [46], Alexnet [47], and Darknet19 [48]. The frameworks are trained on millions of images from the ImageNet database in a TL configuration. All networks’ pre-trained versions can classify images into 1000 separate classes. The final three layers are fine-tuned to separate the chest radiograph images into three groups: COVID-19 positive and COVID-19 negative, i.e., normal and pneumonia. The image input size of models varied, so we resized the chest radiograph images of the dataset according to the input image requirement of the models. We utilized the same experimental settings, as shown in Table 1, to tune the models as we did for the proposed model. For this experiment, we used all the 15,153 radiograph images (3616 images of COVID-19 patients, 1345 Viral Pneumonia radiographs, and 10,192 radiographs of normal individuals) of the dataset named COVID-19 Radiography Database. The 12,123 radiographs (2893 COVID-19 radiographs, 1076 Viral Pneumonia radiographs, and 8154 radiographs of normal individuals) were used for training, whereas the remaining 3030 radiographs (723 COVID-19 radiographs, 269 Viral Pneumonia radiographs, and 2038 radiographs of normal individuals) were used for testing. From the findings shown in Table 13, it is obvious that SqueezeNet achieved the lowest performance results in terms of all performance metrics. DarkNet19 attained the second-best classification accuracy of 99.67%. It is important to mention that the proposed model achieved the optimal precision, recall, and F-measure of 100%. Based on the results, we noticed that the proposed framework performed better than the other DL frameworks by achieving accuracy, precision, recall, specificity, and F1-scores of 99.98%, 100%, 100%, 99.97%, and 100% for COVID-19 detection using chest radiograph images. The AlexNet model has a lower accuracy than the proposed model since each convolutional layer in AlexNet is followed by the Relu activation function. The Relu sets all values smaller than (x < 0), i.e., negative values, to zero for all neurons with negative values. There is no assurance that all neurons will be activated all of the time, which leads to the dying Relu problem. The model does not learn in this scenario because the optimization algorithm does not work. The dying ReLU problem is problematic because it causes a significant part of the network to become inactive over time. Because the proposed model applies BN after each convolutional layer, the significance of each feature is preserved, even though some features have a higher numerical value than others. As a result, the proposed model will be completely unbiased (to higher-value features). In addition, as compared to a framework that does not use BN, the framework that uses this technique is trained faster and has a higher accuracy.

#### 4.6.2. COVID-19 Detection Using CT Scans Comparison with State-of-the-Art Deep Learning Models

This experiment aims to evaluate the usefulness of the proposed methos for COVID-19 identification over the contemporary DL frameworks using CT scans. For this purpose, we compared the detection performance of our framework with, i.e., Squeezenet [46], GoogleNet [49], MobileNetv2 [50], and DenseNet [51]. The frameworks are trained on millions of images from the ImageNet database in a TL configuration. All networks’ pre-trained versions can categorize images into 1000 different classes. The final three layers are fine-tuned to separate the CT scans into COVID-19 positive and COVID-19 negative, i.e., normal. The image input size of models varied, so we resized the CT scan images of the dataset. We utilized the same experimental settings, as shown in Table 1 to tune the models as we did for our model. For this experiment, we used all the 2482 COVID-19 CT scans (1252 COVID-19 CT scans and 1230 normal CT scans) of the dataset named SARS-COV-2 CT scan dataset. We used 1986 CT scans (1002 COVID-19 and 984 normal images) to train the model, whereas the remaining 496 images (250 COVID-19 and 246 normal images) were used for testing. It is included from Table 14 that the proposed approach outperformed other fine-tuned DL frameworks by achieving optimal results in terms of all performance metrics. MobileNetv2 achieved the second-best accuracy of 99.80%, whereas SqueezNet and GoogleNet achieved the minimum accuracy of 99.60%. SqueezeNet achieved recall and specificity of 100%, whereas GoogleNet achieved a recall of 100%. Both SqueezeNet and GoogleNet achieved the same accuracy and F1-score of 99.60% and recall of 100%. It is important to mention that GoogleNet achieved lower precision and specificity than SqueezeNet. So, we can say that GoogleNet is a worse performing model in terms of precision and specificity compared to Squeezenet in this case. The second-best accuracy is achieved by the Mobilenetv2 model, which employs the concept of depthwise convolution and pointwise convolutions with 1 × 1 convolution to capture the most significant information. A linear bottleneck is utilized between the layers to avoid nonlinearities from causing more information loss. Despite the fact that Densenet201 is deeper than ShuffleNet, the results indicate that ShuffleNet’s efficacy stems from its efficient structure rather than its depth. ShuffleNet is built on pointwise group convolutions and channel shuffle strategies, allowing for more feature map channels and hence more knowledge to be encoded, which is extremely significant for the performance of small networks. It is worth mentioning that, when compared to ShuffleNet, these methods are more computationally intensive. The proposed framework attained the best results because of its capacity to extract more detailed features of CT scans by using small filters of 1 × 1 and 3 × 3.

#### 4.6.3. COVID-19 Five-Class Classification Using ECG Trace Images Comparison with State-of-the-Art Deep Learning Models

This experiment aims to evaluate the usefulness of the proposed frameworks for COVID-19 detection over the contemporary DL frameworks using ECG trace images. The proposed approach achieved the highest accuracy in a five-class classification scheme using ECG trace images. We compared the proposed method with contemporary models just for COVID-19 five-class classification rather than binary and three-class classification. For this purpose, we compared the classification performance of the framework with, i.e., Squeezenet [46], GoogleNet [49], DarkNet19 [48], Inceptionv3 [52], and Resnet101 [53]. All TL frameworks are trained on millions of images from the ImageNet database in a TL configuration. All networks’ pre-trained versions can categorize images into 1000 different classes. The final three layers are fine-tuned to separate the ECG trace images into three groups: COVID-19, Normal, and other CVDs. The image input size of models varied, so we resized the ECG images of the dataset. We utilized the same experimental settings (shown in Table 1) to tune the models as we did for our model. For this purpose, we used all the 1937 ECG images (250 COVID-19, 859 normal, 77 MI, 548 AHB, and 203 RMI) of the dataset. In comparison, 1548 images (200 COVID-19, 687 normal, 61 MI, 438 AHB, and 162 RMI) were used for training and the remaining 389 images (50 COVID-19, 172 normal, 16 MI, 110 AHB, and 41 RMI) for testing. The results shown in Table 15 depicts that the proposed method outperformed other fine-tuned DL models by achieving the highest results in terms of all performance metrics. GoogleNet achieved the second-best accuracy of 99.28%, whereas SqueezNet achieved the minimum accuracy of 97.83%. It is important to mention that GoogleNet achieved lower precision and specificity than DarkNet19. So, we can say that GoogleNet is a worse performing model in terms of precision and specificity compared to DarkNet19 in this case. It is to be noted that SqueezeNet, DarkNet19, GoogleNet, and ShuffleNet are 18, 19, 22, and 50 layers deep, respectively. The results show that accuracy increases with model depth, as deeper CNN’s collect more complicated features and improve the model’s classification performance. However, Resnet101 (101 layers deep), a strong structure, led to worse results (accuracy of 90.15%) than the proposed approach. The Resnet101 and GoogleNet uses a relu activation function after each convolutional layer that makes a large portion of the network inactive (as relu makes all negative values zero). In contrast, the ShuffleNet does not use the Relu activation function after depthwise convolutions. Additionally, ShuffleNet uses 1 × 1 convolutions that extract more detailed and distinctive features than Resnet variants that use 3 × 3 and 7 × 7 convolutions. Because of its ability to extract more complex features for COVID-19 identification using ECG trace images, the proposed COVID identification model achieved comparatively better results. These results show the superiority of the proposed model over other contemporary models.

### 4.7. Comparison to the State-of-the-Art Approaches

We planned a multi-stage experiment to compare the suggested and existing state-of-the-art COVID-19 identification approaches to verify the proposed model’s superiority over existing methodologies. We compared the proposed approach to the most recent DL frameworks and presented the results in Table 16.

In the first stage, we compared the proposed model with the contemporary COVID-19 detection methods [11,12,54] for three-class classification using chest radiographs. The results are presented in Table 16 (row 1 and row 2). Kumar et al. [11] presented and analyzed the performance of SARS-Net, to detect irregularities in a patient’s chest radiographs for the presence of COVID-19 infection, SARS-Net merged Graph Convolutional Networks with CNN. On the validation set, the suggested model was found to have an accuracy of 97.60% and a sensitivity of 92.90%. To detect COVID-19 from chest radiograph images, the authors developed an inverted bell-curve-based ensemble of DL frameworks in [12]. For this purpose, the pre-trained models were first retrained with radiograph datasets using a TL method and integrated with the suggested inverted bell curve weighted ensemble approach, which assigns a weight to each classifier’s output and performs a weighted average of those outputs to get the final prediction. Two datasets were used to test the suggested method: the COVID-19 Radiography Database and the IEEE COVID Chest X-ray dataset. In the first dataset and the other dataset, the suggested technique achieved 99.66% and 99.84%, respectively. 

In the second stage, we compared the proposed approach with the contemporary COVID-19 identification approaches [55,56] using CT scan images, and the findings are shown in Table 16 (rows 5 and 6). For detecting COVID-19 from CT scan images, Basu et al. [55] suggested an end-to-end system with feature extraction followed by feature selection. Three CNNs were used to extract feature information. For feature selection, they merged Harmony Search (HS), a meta-heuristic optimization technique, with Adaptive *β*-Hill Climbing (A*β*HC), a local search approach. The technique achieved the best accuracy ratings of 98.87% on the SARS-COV-2 CT scan dataset. Alquzi et al. [56] proposed a classification approach using CT scans and machine learning to diagnose patients with COVID-19 infection. The system was built using the EfficientNet model. The authors altered the Efficientnet-B3 model by deleting the top layer and replacing it with two branches with different layers. The proposed approach was tested on the SARS-CoV-2 CT dataset and achieved an accuracy of 99%.

Finally, in the third stage, we compared the proposed classifier with the contemporary COVID-19 detection approaches [39] using ECG trace images, and the results are shown in Table 16 (rows 7 and 9).

In the third case, the proposed model performed the best and achieved an accuracy of 99.37% for five-class classification and 98.96% for three-class classification. In contrast, Rahman et al. [39] produced an accuracy of 97.83% for five-class classification and achieved an accuracy of 97.36% for three-class classification. It is significant to mention that the research methods compared with the proposed framework detect COVID-19 using one image type, i.e., chest radiographs, CT scans, or ECG trace images. To the best of our knowledge, no research study uses three types of images data to detect COVID-19. As the proposed method satisfactorily identifies and detects COVID-19 using three types of images data, we can conclude that the proposed approach is more effective for COVID-19 detection and classification.

## 5. Discussions

This study presents an effective COVID-19 detection and COVID-19 and other diseases (such as CVDs, Pneumonia, and Lung Obesity) classification approach using the TL approach by employing three types of images. We performed extensive classification (binary and multiclass) experiments with the proposed approach using each type of image (chest radiographs, CT scans, and ECG trace images). We evaluated the proposed approach with the baseline datasets and achieved a higher accuracy than state-of-the-art approaches. We achieved an accuracy of 99.82%, 99.98%, and 99.78% for binary, three-class, and four-class classifications, respectively, using chest radiographs. Furthermore, we achieved the optimal accuracy of 100% in the case of binary classification using CT scans. Finally, we achieved an accuracy of 99.10%, 98.96%, and 99.37% in the case of binary, three-class, and five-class classification schemes using ECG images.

This research study will aid in the speedier computer-aided detection of COVID-19 and other cardiac disorders. The proposed approach employs ECG trace images collected by cellphones and are commonly accessible in nations with limited resources and budgets. Chest radiographs enable community hospitals and local health centers to do investigations in a short amount of time. The suggested method’s key advantage is that it outperforms current state-of-the-art techniques without necessitating a segmentation process. The proposed approach will be valuable in supporting doctors in formulating clinical judgments to determine COVID-19 as soon as possible and strict the flow of contagious and viral diseases, such as Covid-19. Furthermore, if this method is used in clinics, the number of treatments will decrease, and it will help to limit doctor involvement and save countries’ health systems from collapsing.

Even though the proposed approach achieved promising results, we still discovered some limitations and recommendations for future work. In this study, the proposed approach could not categorize the numerous stages of COVID-19 infection, such as pre-symptomatic, asymptomatic, moderate, and severe, due to a lack of datasets that could be utilized to examine the severity level of COVID-19 infection. The proposed approach does not identify how well the system detects COVID-19 using other imaging modalities, such as lung ultrasound and lung PET (positron emission tomography) scans. The proposed approach is currently trained and validated with comparatively limited data set instances. However, the number of training data significantly influences the performance and effectiveness of DL-based frameworks; therefore, the proposed approach may also be expanded by employing a large-scale dataset containing millions of images. We consistently split images data into an 80% training set and a 20% test set in the proposed approach. It is possible, though, that alternative splits will provide different outcomes.

In the future, we intend to perform experiments using comparatively large-scale datasets of chest radiographs, CT scans, and ECG trace images to evaluate the models’ identification and classification ability by testing them on a variety of large-scale datasets from various sources and with images obtained by multiple machines. We intend to use the same approach to categorize the numerous stages of COVID-19 infection, such as pre-symptomatic, asymptomatic, moderate, severe, etc. Additionally, we will use the proposed method in other COVID-19 datasets or other medical datasets with CT scans or chest radiographs. Another future direction is to employ different image segmentation and pre-processing approaches to remove noise, i.e., darkness, low contrast, damaged pixels, or occlusion from the input images, particularly, CT-scan, to improve the classifier results. Additionally, due to the emergence of the internet, many countries’ health care institutions are using many smart and intelligent electronic devices to combat diseases and obtain pivotal information about their growth. Furthermore, IoT and blockchain technologies are helping medical professionals to obtain essential insights and information about patients’ symptoms and behaviors. Similarly, medical physicians utilize IoT-enabled devices to remotely monitor patients, assuming that COVID-19 spreads faster than the average viral disease. More recently, Internet of Medical Things (IoMT) applications and devices have been heavily used in the medical domain to gather, analyze, and transmit healthcare-related information. Therefore, IoT-based medical devices can improve and enhance the diagnosis and detection process of various infectious diseases, which is essential in the case of COVID-19. Furthermore, IoT-enabled medical devices can capture the body temperature, gather samples and information from the infected patients, and minimize human intervention to reduce virus spread. In addition, during the quarantine period of COVID-19 positive patients, IoT medical devices can monitor infected patients remotely to prevent the spread of the virus. Therefore, it is necessary to propose machine and deep learning approaches based on blockchain and IoT technologies because both significantly leverage the global healthcare industry to timely detect and identify COVID-19 from the data generated by these devices. Furthermore, to generalize the proposed approach in detecting other important medical diseases [58,59], we aim to validate the performance of the proposed approach by training and testing it on the identification of brain tumors [60,61], pest detection [62], heart diseases [63,64], and mask detection [65], blood diseases [66,67,68].

## 6. Conclusions

This work presents an effective method for automated COVID-19 detection by employing the ShuffleNet DL model in a TL setup. The ShuffleNet framework was chosen for its proven effectiveness in image detection and classification tasks. The presented DL-based model reliably and accurately detects COVID-19 using three types of images data, i.e., chest radiographs, CT scans, and ECG trace images. Moreover, we have validated the robustness and generalizability of the proposed model on three publicly available datasets. The accuracy of 99.98% for COVID-19 detection in the case of three-class classification using chest radiographs, optimal accuracy of 100% for COVID-19 detection using CT scans, and 99.37 for five-class classification using ECG trace images have verified the effectiveness of our proposed model over the contemporary methods. The optimal accuracy of 100% for COVID-19 detection using CT scans and the accuracy gain of 1.54% (in the case of five-class classification using ECG trace images) from the previous approach, which utilized ECG images for the first time, has a major contribution to improving the COVID-19 prediction rate in early stages. Experimental results demonstrate that the proposed framework outperforms the existing COVID-19 detection and classification approaches.

## Figures and Tables

**Figure 1 diagnostics-13-00162-f001:**
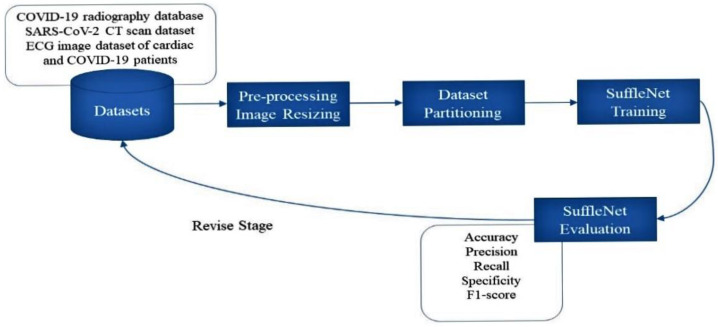
Overview of the proposed method.

**Figure 2 diagnostics-13-00162-f002:**
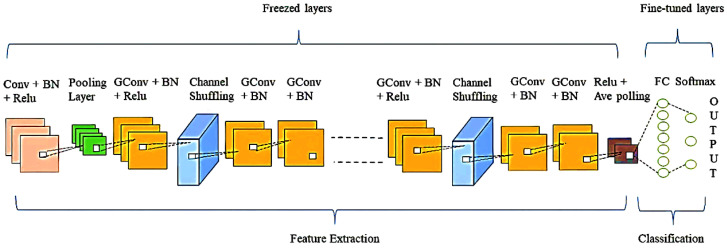
ShuffleNet convolution neural network architecture of the proposed framework, the GConv represents group convolutions.

**Figure 3 diagnostics-13-00162-f003:**
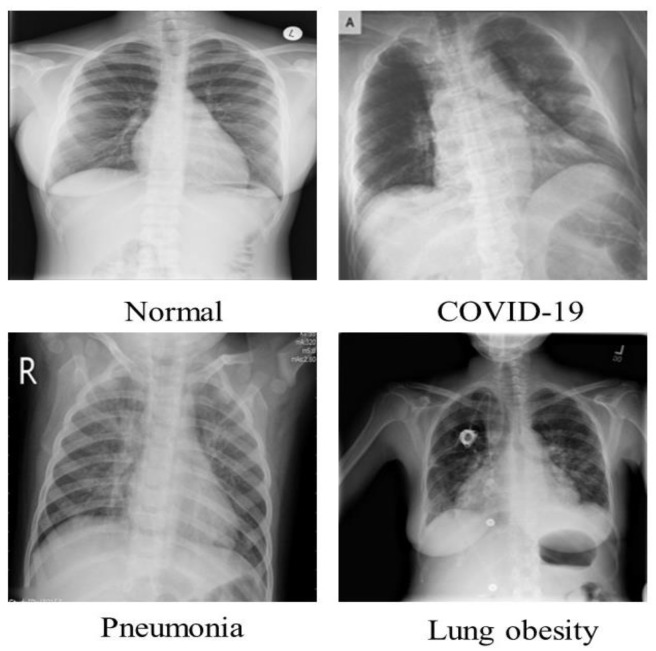
Normal, COVID-19, Pneumonia and Lung obesity images samples of the dataset.

**Figure 4 diagnostics-13-00162-f004:**
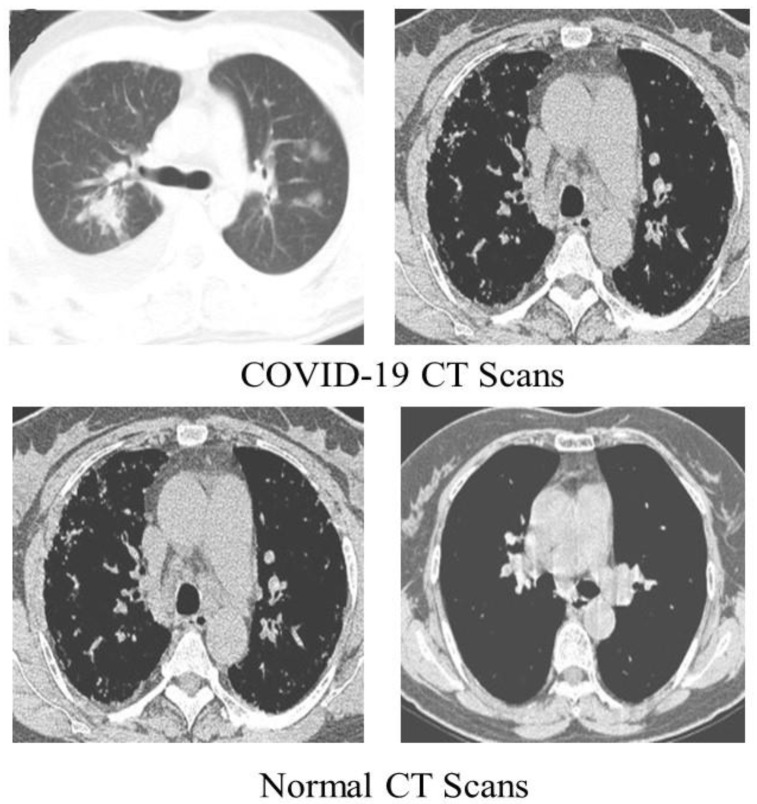
COVID-19 (**first row**) Normal CT scan (**second row**) samples of the dataset.

**Figure 5 diagnostics-13-00162-f005:**
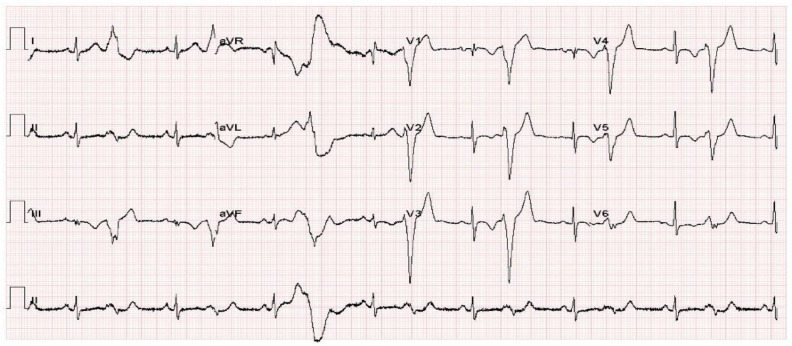
Sample COVID-19 ECG trace image from the dataset, the time is represented by the horizontal axis; each time step lasts 0.04 s and is indicated by a vertical line. The vertical axis shows signal magnitudes in millivolts.

**Figure 6 diagnostics-13-00162-f006:**
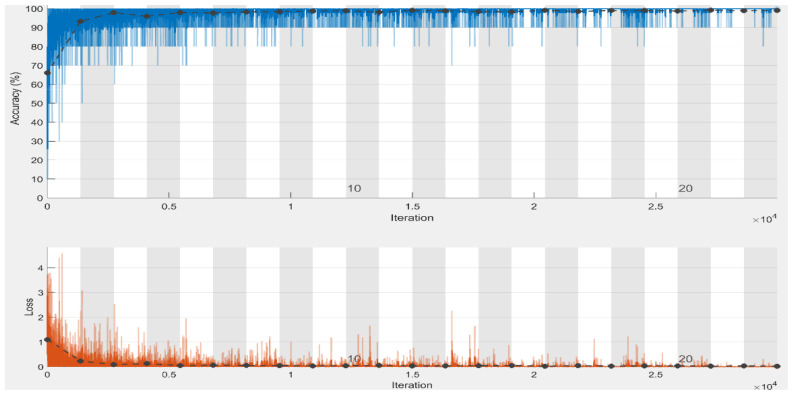
Training and validation accuracy and loss plots of three-class classification (black line shows the testing accuracy and loss) using chest radiographs dataset.

**Figure 7 diagnostics-13-00162-f007:**
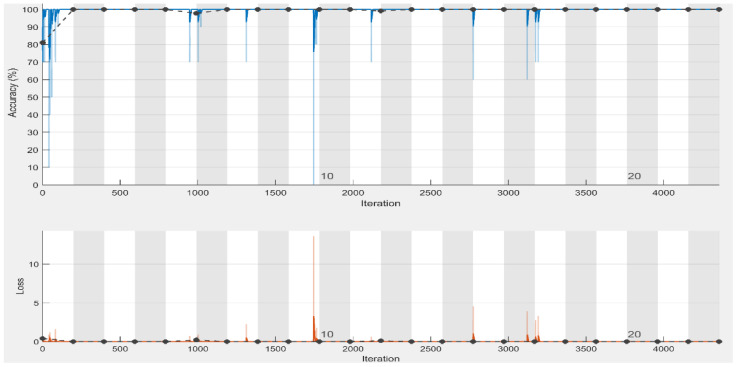
Training and validation accuracy and loss plots of binary classification (black line shows the validation accuracy and loss) using CT scan images.

**Figure 8 diagnostics-13-00162-f008:**
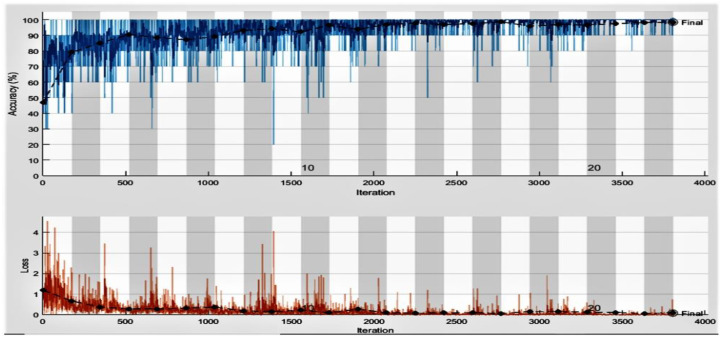
Training and testing accuracy and loss plots of five-class classification (black line shows the testing accuracy and loss) using ECG trace images dataset.

**Table 1 diagnostics-13-00162-t001:** Hyper-parameters values.

Parameter	Value
Optimization algorithm	SGDM
Learning rate	0.001
Shuffle	Every epoch
Maximum epochs	22
Validation frequency	30
Mini batch size	10
Activation function	Relu
Verbose	false

**Table 2 diagnostics-13-00162-t002:** Datasets details.

Images Type	Dataset	Images Collections					Format	Bit Depth
Chest Radiographs	Chest Radiography Database	COVID	Normal	Pneumonia	Lung Obesity		PNG	8
		3616	10,192	1345	6012			
CT Scan	SARS-CoV-2 CT scan dataset	COVID		Non-CCOVID			PNG	24, 32
		1252		1230				
ECG trace	ECG image dataset of cardiac and COVID-19 patients	COVID	Normal	RMI	AHB	MI	JPG	24
		250	859	203	548	77		

**Table 3 diagnostics-13-00162-t003:** Confusion matrix attained by ShuffleNet in binary classification scheme using chest radiographs.

**True Class**	**Predicted Class**		
	Class	COVID-19	Normal
	COVID-19	221	2
	Normal	3	2035

**Table 4 diagnostics-13-00162-t004:** Confusion matrix attained by ShuffleNet in three-class classification using chest radiographs.

**True Class**	**Predicted Class**			
	Class	COVID-19	Normal	Pneumonia
	COVID-19	722	1	0
	Normal	0	2038	0
	Pneumonia	0	0	269

**Table 5 diagnostics-13-00162-t005:** Confusion matrix attained by ShuffleNet in four-class classification using chest radiographs.

**True Class**	**Predicted Class**				
	Class	COVID-19	Lung obesity	Normal	Pneumonia
	COVID-19	723	0	0	0
	Lung obesity	0	1197	5	0
	Normal	1	13	2024	0
	Pneumonia	0	0	0	269

**Table 6 diagnostics-13-00162-t006:** Performance evaluation on COVID-19 Radiography Database multi-class classification.

Classification Scheme	Accuracy	Precision	Recall	Specificity	F1_Score
COVID-19, Normal, and Pneumonia	99.98	100	100	99.97	100
COVID-19, Normal, Pneumonia, and Lung Obesity	99.78	99.75	99.75	99.93	99.75

**Table 7 diagnostics-13-00162-t007:** Confusion matrix attained by ShuffleNet in binary classification scheme using CT scans.

**True Class**	**Predicted Class**		
	Class	COVID-19	Normal
	COVID-19	250	0
	Normal	0	246

**Table 8 diagnostics-13-00162-t008:** Evaluation on ECG Images dataset of Cardiac and COVID-19 Patients binary classification.

Classification Scheme	Accuracy	Precision	Recall	Specificity	F1_Score
COVID-19 and Normal	99.10	96.00	1.00	98.85	97.96
COVID-19 and myocardial infarction	96.92	96.00	1.00	88.24	97.96
COVID-19 and abnormal heartbeat	98.74	96.00	1.00	98.20	97.96
COVID-19 and recovered myocardial infarction	97.80	96.00	1.00	95.35	97.96

**Table 9 diagnostics-13-00162-t009:** Confusion matrix attained by ShuffleNet in three-class classification scheme using ECG images.

**True Class**	**Predicted Class**			
	Class	COVID-19	Normal	Other diseases
	COVID-19	48	0	2
	Normal	0	170	2
	Other disease	0	2	163

**Table 10 diagnostics-13-00162-t010:** Confusion matrix attained by ShuffleNet in five-class classification scheme using ECG images.

**True Class**	**Predicted Class**					
	Class	COVID-19	MI	RMI	AHB	Normal
	COVID-19	48	0	0	2	0
	MI	0	14		1	0
	RMI	0	0	40	0	1
	AHB	0	1	0	108	
	Normal	0	0	0	1	171

**Table 11 diagnostics-13-00162-t011:** Performance evaluation on ECG Images dataset of Cardiac and COVID-19 Patients multiclass classification.

Classification Scheme	Accuracy	Precision	Recall	Specificity	F1_Score
COVID-19, Normal, and others	98.96	98.00	99.00	99.09	98.33
COVID-19, Normal, MI, AHB and MI	99.37	97.0	97.60	99.12	97.29

**Table 12 diagnostics-13-00162-t012:** Hybrid approach results.

Approach	Classification Scheme	Dataset	Accuracy	Precision	Recall	Specificity	F1_Score	Time Elapsed
Shufflenet and SVM	Binary classification	CT scan dataset	91.33	91.5	91.5	93.56	91.5	2 min 32 s
Proposed method	Binary classification	CT scan dataset	100	100	100	100	100	329 min
Shufflenet and SVM	Three-class classification	Radiograps Database	96.34	92.66	94.33	96.56	93.59	15 min 8 s
Proposed method	Three-class classification	Radiograps Database	99.98	100	100	99.97	100	1460 min
Shufflenet and SVM	Five-class classification	ECG Images dataset	89.74	67	68.2	82.19	67.49	3 min 13 s
Proposed method	Five-class classification	ECG Images dataset	99.37	97.0	97.60	99.12	97.29	193 min

**Table 13 diagnostics-13-00162-t013:** COVID-19 three-class classification using chest radiographs comparison with state-of-the-art frameworks.

Model	Accuracy	Precision	Recall	Specificity	F1_Score
Squeezenet	98.29	98.33	94.67	95.02	96.46
Alexnet	98.50	98.66	95.00	95.85	96.79
Darknet19	99.67	99.66	99.00	99.43	99.32
Proposed model	99.98	100	100	99.97	100

**Table 14 diagnostics-13-00162-t014:** COVID-19 binary classification using CT scan images comparison with state-of-the-art models.

Model	Accuracy	Precision	Recall	Specificity	F1_Score
Squeezenet	99.60	99.21	100	100	99.60
Googlenet	99.60	99.20	100	99.19	99.60
Mobilenetv2	99.80	100	99.60	100	99.80
Densenet201	99.40	99.5	99.5	100	99.5
Proposed model	100	100	100	100	100

**Table 15 diagnostics-13-00162-t015:** COVID-19 five-class classification using ECG trace images comparison with state-of-the-art models.

Model	Accuracy	Precision	Recall	Specificity	F1_Score
Squeezenet	97.83	89.00	94.40	95.01	91.62
Googlenet	99.28	96.80	97.60	98.52	97.19
Darknet19	99.17	97.8	95	99.10	96.38
Resnet101	90.15	87.33	88.33	88.67	87.82
Mobilenetv2	91.71	74.2	75.2	89.24	74.70
Inceptionv3	80.61	78.31	79.23	79.2	78.76
Proposed model	99.37	97.0	97.60	99.12	97.29

**Table 16 diagnostics-13-00162-t016:** Accuracy comparison with state-of-the-art models in the literature.

S. No	Work	Classification Scheme	Method	Date	Accuracy
Comparison with previous COVID-19 detection approaches using COVID-19 Radiography Database					
1	Sanida et al. [54]	COVID-19, Normal, and pneumonia	Light weight CNN	2022	95.80
2	Kumar et al. [11]	COVID-19, Normal, and pneumonia	SARS-Net	2022	97.60%
3	Paul et al. [12]	COVID-19, Normal, and pneumonia	Ensemble method	2022	99.66%
4	Proposed approach	COVID-19, Normal, and pneumonia	ShuffleNet	2022	99.98
Comparison with previous COVID-19 detection approaches using SARS-COV-2 CT Scan Dataset					
5	Basu et al. [55]	COVID-19 and Normal	Two-stage framework	2022	98.87%
6	Alquzi et al. [56]	COVID-19 and Normal	Efficientnet-B3	2022	99.0%
7	Dutta et al. [57]	COVID-19 and Normal	EDLFM-SI	2022	96.25
8	Proposed approach	COVID-19 and Normal	ShuffleNet	2022	100%
Comparison with previous COVID-19 detection approaches using ECG Images of Cardiac and COVID-19 Patients					
9	Rahman et al. [39]	COVID-19, Normal, and other CVDs	COV-ECGNET	2021	97.36%
10	Proposed approach	COVID-19, Normal, and other CVDs	ShuffleNet	2022	98.96
11	Rahman et al. [39]	Normal, COVID-19, MI, AHB, and RMI	COV-ECGNET	2021	97.83%
12	Proposed approach	Normal, COVID-19, MI, AHB, and RMI	ShuffleNet	2022	99.37

## Data Availability

The datasets used in this investigation are available on request from the corresponding author.

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
