# Peer review of "A Holistic Approach to Identify and Classify COVID-19 from Chest Radiographs, ECG, and CT-Scan Images Using ShuffleNet Convolutional Neural Network"

_diagnostics, 2023, doi:10.3390/diagnostics13010162_

Round 1

Reviewer 1 Report

I would suggest the auhtors in this paragraph to mention data from worldwide not just Pakistan-. According to the Pakistani 61 government, 1,518,083 cases of COVID-19 have been reported in Pakistan, with 30,304 fa- 62 talities and 1,469,930 recoveries [5].-

Please provide bibliography for this paragraph-To reliably and automatically detect (identify or predict) COVID-19 in its early 73 phases, various medical imaging techniques, such as chest radiographs, ECG trace im- 74 ages, and CT-scan have been used. Chest radiographs, often known as X-rays, are images 75 of the inside of the chest that are utilized to examine chest problems. ECG trace images 76 are line graphs that depict variations in the heart's electrical behavior over time. On the 77 other hand, a chest CT scan employs an x-ray scanner to produce a sequence of high- 78 resolution images of locations inside the chest. Medical professionals value and prefer 79 chest radiograph images more because they can be easily accessed from radiology depart- 80 ments. Chest radiograph images, according to radiologists, aid in the clear understanding 81 of chest pathology. Also, the ECG trace images can easily be taken and gathered by mobile 82 phones and are quickly accessible technologies in nations with limited resources and 83 budgets.-

The conclusion has to be restructured. The conclusion should be just one paragraph and should not include bibliography.

Author Response

Dear Reviewer,

Thank you for allowing a resubmission of our manuscript, 
with an opportunity to address the interesting and beneficial comments and improve the overall quality of the paper by incorporating all the comments. Also, we highlighted the changes in the revised manuscript to make it visible and prominent.

We attached a point-by-point response to the comments, (b) an updated manuscript.

Reviewer 2 Report

The paper is written. The authors proposed a DL-based framework to identify COVID-19-positive cases by examining chest radiographs, CT scans, and ECG 111 trace images. The proposed approach is based on a filter-feature extraction, which can help attain the most extraordinary classification performance. However, I believe some issues need to be addressed before publication.

1.          The manuscript title is way too general.

2.          Some language issues need to be checked

3.          The figures’ resolution can be improved especially figure 1 and 2.

4.          The comparison was detailed in an excellent analytical way; however, I think the comparison and discussion should be separated to provide better insights.

Author Response

(The authors gave the same response as above.)
